# Using Patient-Reported Outcome Measures (PROMs) to promote quality of care and safety in the management of patients with Advanced Chronic Kidney disease (PRO-trACK project): a mixed-methods project protocol

Olalekan Lee Aiyegbusi,[1,2] Derek Kyte,[1,2] Paul Cockwell,[1,3] Tom Marshall,[1,2] Mary Dutton,[3] Anita Slade,[1,2] Neil Marklew,[4] Gary Price,[4] Rav Verdi,[4] Judi Waters,[4] Keeley Sharpe,[4] Melanie Calvert[1,2]

For numbered affiliations see end of article.

**Correspondence to**
Derek Kyte; d.g.kyte@bham.ac.uk

## ABSTRACT

**Introduction** Advanced chronic kidney disease (CKD) has a major effect on the quality of life and health status of patients and requires accurate and responsive management. The use of electronic patient-reported outcome measures (ePROMs) could assist patients with advanced pre-dialysis CKD, and the clinicians responsible for their care, by identifying important changes in symptom burden in real time. We report the protocol for 'Using Patient-Reported Outcome measures (PROMs) to promote quality of care and safety in the management of patients with Advanced Chronic Kidney Disease' (PRO-trACK) project, which will explore the feasibility and validity of an ePROM system for use in patients with advanced CKD.

**Methods and analysis** The project will use a mixed-methods approach in three studies: (1) usability testing of the ePROM system involving up to 30 patients and focusing on acceptability and technical performance/stability; (2) ascertaining the views of patient and clinician stakeholders on the optimal use and administration of the CKD ePROM system—this will involve qualitative face-to-face/telephone interviewing with up to 30 patients or until saturation is achieved, focus groups with up to 15 clinical staff, management and IT team members; (3) psychometric assessment of the system, within a cohort of at least 180 patients with advanced CKD, to establish the measurement properties of the ePROM.

**Ethics and dissemination** This project was approved by the West Midlands Edgbaston Research Ethics Committee (Reference 17/WM/0010) and received Health Research Authority (HRA) approval on 24 February 2017. The findings from this project will be provided to clinicians at the Department of Renal Medicine, Queen Elizabeth Hospitals, Birmingham (QEHB), NHS England, presented at conferences and to the Kidney Patients' Association, British Kidney Patient Association and the British Renal Society. Articles based on the findings will be written and submitted for publication in peer-reviewed journals.

## Strengths and limitations of this study

► While there is evidence to support the use of electronic patient-reported outcome measures (ePROMs) in the management of other conditions, notably cancer, the evidence for the use of ePROMs in the management of patients with chronic kidney disease (CKD) is currently limited. The PRO-trACK project will help fill this evidence gap.

► By using a mixed-methods approach, the project will provide a rigorous exploration of the acceptability, validity and feasibility of the ePROM system for the management of patients with CKD.

► This project will only involve patients with CKD stages 4 and 5 and patients on dialysis for ≤6 months. This is because the ePROM system is presently intended for patients with advanced CKD stages 4 and 5 who we hypothesise are likely to derive the most benefit.

## INTRODUCTION

Chronic kidney disease (CKD) refers to a number of disorders affecting the structure and function of kidneys.[1] The definition of CKD is based on sustained reduction in renal function (ie, 'estimated glomerular filtration rate (eGFR) <60 mL/min per 1.73 m$^2$ for 3 months or more') and/or evidence of structural or functional abnormalities of the kidneys regardless of clinical diagnosis.[1 2] CKD is associated with other long-term conditions such as hypertension, cardiovascular diseases and diabetes that will increase the risk of ill health both when present singly and when associated with other long-term conditions.

CKD causes clinical signs and symptoms, particularly when the disease is advanced.[3]

The most commonly experienced are fatigue, drowsiness, pain, pruritus and dry skin.[4] These symptoms often occur concurrently and may negatively affect patients' daily activities and their physical, emotional and psychological well-being,[5] therefore impacting on the quality of life (QOL) of those affected, particularly as the disease progresses towards end-stage renal disease.[6 7]

The symptoms of CKD progression can be monitored using self-completed questionnaires known as patient-reported outcome measures (PROMs), which capture information about health status from patients' own point of view.[8] Although commonly administered in paper format, PROMs can be completed as electronic patient-reported outcome measures (ePROMs) using multiple digital platforms. This makes it possible to remotely monitor patients and generate 'real time' data about patient symptoms and QOL. As patients with advanced CKD are at risk of deteriorating rapidly and developing cardiovascular complications,[9] the use of an ePROM system may help clinicians detect deterioration of symptoms and assist with the tailoring of treatment to the needs of each patient.[10–12] Health-related issues that matter to patients may also be identified using ePROM data, and this could potentially facilitate communication and shared decision-making between patients and their clinicians.[13–15] In stable patients, the use of ePROMs may reduce the occurrence of unnecessary clinical appointments.[12]

In Denmark, the WestChronic ePROM System has been successfully implemented for tailoring the care of various patient groups,[12] while in the UK, patients with cancer have been successfully monitored for the side effects of chemotherapy using the ePROM Advanced Symptom Management System.[16] However, there is limited information on the use of ePROMs in the management of adult patients with CKD in a routine clinical setting. Therefore, the aim of the project is to explore the feasibility and validity of an ePROM system for monitoring and assisting with the individual management of patients with advanced CKD.

## Questionnaire selection and the ePROM system

Selection of measures was informed by (1) a systematic review of measurement properties of PROMs used in patients with CKD[17] and (2) feedback from the patient advisory group (PAG). The systematic review found evidence to support the use of the Kidney Disease Quality of Life-Short Form (KDQOL-SF) and Kidney Disease Quality of Life-36 (KDQOL-36). However, these two measures were validated by very few studies in our population of interest (stages 4 and 5 CKD). The review also identified the Integrated Patient Outcome Scale-Renal (IPOS-Renal), which is currently undergoing validation through use in a number of renal units in the UK.

A PAG met prior to commencing the project and considered the acceptability, burdensomeness and relevance of the KDQOL-SF, KDQOL-36 and the IPOS-Renal for the target CKD group. The PAG members chose the KDQOL-36 and IPOS-Renal as they were brief and easy to understand. Therefore, the decision was made to validate the electronic versions of these in the pre-dialysis population (stages 4 and 5). The EuroQol Five-Dimension Five-Level (EQ-5D-5L) questionnaire will be used as a comparison measure for this

| Table 1 | Description of questionnaires |
| --- | --- |
| **Measure** | **Description** |
| Kidney Disease Quality of Life-36 (KDQOL-36) | A 36-item health-related quality of life (HRQOL) measure designed for use in patients with kidney disease undergoing dialysis. Derived from the Kidney Disease Quality of Life-Short Form.[88] There are three disease-specific dimensions, namely (1) symptoms and problems (six items), (2) burden of kidney disease (four items) and (3) effects of kidney disease (eight items). It also includes two summary scales derived from the generic Short Form (SF)-12, namely (1) the physical component summary (six items) and (2) the mental component summary (six items).[89] Response options vary for the items and range from 2 to 6. Questions 1, 8 and 12–36 have five response options; questions 2 and 3 have three response options; questions 4–7 have two response options; and questions 9–11 have six response options each. Total and subscale scores may be calculated using the KDQOL-36 scoring program. The raw scores for each item are converted linearly to a 0 to 100 range with higher scores indicating better HRQOL.[88] |
| Integrated Patient Outcome Scale-Renal | A symptom-specific measure with 11 questions for use with patients with advanced kidney disease to assess their care needs. The questions relate to common symptoms patients with renal disease experience plus additional items such as information needs, practical issues and family anxiety.[90] The first question has a free text response format. Questions 2 to 9 have five response options while questions 10 and 11 have three response options each. The measure is currently being validated by researchers at the Department of Palliative Care, Policy and Rehabilitation at King's College London. Dimensions yet to be ascertained. |
| EuroQol Five-Dimension Five-Level questionnaire | A generic utility measure with a self-classifier and a visual analogue scale, which can be used to measure health status.[91 92] The self-classifier includes five dimensions: (1) mobility, (2) self-care, (3) usual activities, (4) pain/discomfort and (5) anxiety/depression. This version of the measure has five levels of severity (response options) for each dimension. It is possible to describe 3125 different health states between 0 (dead) and 1 (perfect health).[91 92] |

validation study. See table 1 for a brief description of these three measures.

The KDQOL-36 and the IPOS-Renal are free to use without charge as long as the developers are appropriately acknowledged and cited. The EQ-5D-5L requires prior written consent and payment of licensing fees (if applicable). A licence will be obtained for this project.

The PRO-trACK project will consist of three studies, namely (1) usability testing with patients, (2) qualitative study with patients and clinicians, and (3) validation study with patients.

While the usability testing and qualitative interviews are related, they are distinct studies. The usability testing will focus on the actual experience of patients as they test the ePROM system while the qualitative interviews will explore the broader opinions of patients on the use of ePROMs in the NHS.

The content validation of the KDQOL-36 and the IPOS-Renal by patients and clinicians during the qualitative study as well as the results of the validation study will assist the research team with the final decision on which measure to take forward for the final version of the ePROM system.

The ePROM system will be designed as an electronic method of allowing patients with CKD to remotely self-report their symptoms and QOL using a digital platform that is convenient to them (PC, tablet, smartphone, telephone voice recognition or scanned paper copy), providing important patient-centred data to the patients' clinical team.

The ePROM system will be accessed via the secure electronic patient portal developed by the University Hospitals Birmingham NHS Foundation Trust, known as 'myhealth@QEHB' (see figure 1).[18] myhealth@QEHB currently has 14 000 patient users and was awarded the prestigious E- Health Insider award in 2014.[19] Around 1200 patients with renal disease are currently signed up for myhealth@QEHB. At the moment, this is a voluntary system.

## METHODS AND ANALYSIS
### Project design
In this section, the project setting and eligibility criteria for participants will be described first as this will be the same for the three studies. Aspects of research methods specific to each study will be subsequently discussed separately.

### Project setting
Queen Elizabeth Hospitals, Birmingham (QEHB) will be the host site for this project. Clinical staff at the Renal Unit, QEHB and academic researchers at the Centre for Patient-Reported Outcomes Research, University of Birmingham will be responsible for the conduct and management of the project. The nephrology service comprises 21 consultants, 30 junior doctors, and 20 nurses and allied health professionals in the CKD team.

### Project participants
#### Patient participants
For the project, we will recruit adult patients with advanced CKD stages 4 and 5 under the care of the renal services at QEHB, with eGFR $\leq 20\,mL/min/1.73\,m^2$ who have been counselled about treatment modalities for end-stage renal disease (ESRD). In addition, using the renal risk calculator, they must have a >20% projected risk of progressing to ESRD and requiring renal replacement therapy (RRT) or an eGFR $<10\,mL/min/1.73\,m^2$ within 2 years.[20] The renal risk calculator is a model designed to use routinely collected laboratory tests to predict the progression of patients with CKD stages 3 to 5 to kidney failure.[20]

We have selected this group of patients as our main target for this project because, even though they have not yet reached ESRD, they are likely to have high symptom

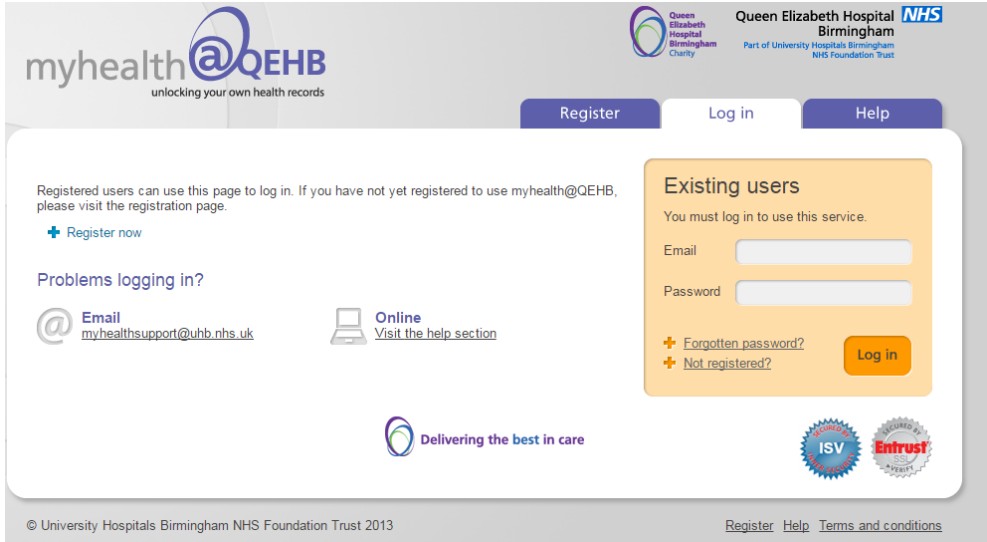

**Figure 1** Screenshot of the myhealth@QEHB login page.

burden and a high risk of rapid clinical deterioration to renal failure. We hypothesise this group of patients are likely to derive the most benefit from an ePROM system. The research team also is working on a related project focused on dialysis patients.

Patients who have commenced dialysis within 6 months will also be eligible to participate as we hypothesise they will be able to recall their symptoms and medical needs pre-dialysis. Participants will be required to converse in everyday English and provide informed consent.

Patients who have a recent history of acute kidney injury within the last 3 months, a comorbidity with a high level of symptoms or terminal illness likely to lead to the death within 6 months of participation will be excluded from the study.

We will aim to recruit different sets of eligible patients for each study in order to minimise participant burden. However, if patients voluntarily express an interest, they will be allowed to participate in more than one study as long as the renal team is satisfied with their health status.

Efforts will be made to recruit up to 30% of the study participants from minority ethnic groups to reflect the ethnic diversity of the patient catchment area.

Although our previous work does not show an influence of socioeconomic status on outcomes for CKD,[21] we will be mindful of sample diversity in relation to sociodemographic variables such age, gender, ethnicity and other relevant sociodemographic factors. We will collect data on participant characteristics to monitor this as recruitment and qualitative data collection progress. Although we appreciate that patient populations and research samples do not always represent such diversity, we will try to employ recruitment strategies that optimise our ability to recruit a diverse patient sample.

### Clinicians and other professional staff
Clinicians who manage patients with CKD at the Renal Unit, QEHB and members of the myHealth team and hospital management staff who provide consent will be recruited for this project.

### Recruitment methods
A member of the renal research team at QEHB will screen patients for eligibility using the electronic screening tools that are used for clinical purposes. This will identify patients who meet the eligibility criteria and the clinics they attend. Eligibility will be confirmed by direct review of the clinical records by the research nurse and a clinician who are members of the renal care team, on the delegated duty log.

Patients will then be approached at clinic by a member of the renal care team and given patient information sheet to read. Further information about the study will be given and their immediate queries will be addressed. They will be contacted no earlier than 48 hours after the clinic visit to ascertain if they wish to participate in the study.

Clinicians and other professional staff will be recruited by the members of the research team and sent information sheets via email.

### Study 1: usability testing
Usability testing will be conducted to evaluate the acceptability of the ePROM system.

#### Study objectives
► To determine whether patients can easily navigate the ePROM system.
► To determine if patients are able to complete the questionnaires successfully on their own and, if not, how much assistance they require.
► To determine the average length of time required to complete an ePROM report.
► To determine the level of satisfaction with the ePROM interface.
► To identify changes that might be required to improve user performance and satisfaction.

#### Data collection
Usability testing refers to the appraisal of a product or service by potential service users and involves the observation of such users completing a task within a predetermined scenario.[22 23]

The scenario for this study will be the self-report of a patient's health status between clinic appointments using an electronic device such as a smartphone, tablet or PC/laptop. Each patient will undergo a single one-to-one session with OLA, the project's chief investigator (CI), and attempt to complete the three electronic questionnaires with as little assistance as possible.

The Concurrent Think Aloud (CTA) and Retrospective Probing (RP) moderating techniques will be used for this study.[24] CTA involves the thinking aloud and vocalisation of participants' thoughts during the session while RP refers to interviewing the participant following the completion of the session.[24] The advantage of combining both techniques is that real-time feedback could be obtained for exploration by the CI afterwards.[22]

The CI will take detailed notes of the participants' comments, actions, non-verbal cues and errors and pass minimal comments to encourage them to think aloud during the sessions.

Qualitative data will be collected in the form of a brief audio-recorded interview at the end of each test. The patients will be questioned based on the notes taken by the CI during the session. They will be encouraged to provide any recommendation to improve user experience.

#### Data analysis
Thematic analysis will be conducted on the qualitative data (see the analysis of patient interviews for more details), which will include the notes taken during the sessions and the transcripts of the post-test interview.

Quantitative data will be summarised using descriptive statistics such as proportions, averages, percentages and rates.[22] Quantitative data will include successful completion rates, error-free rates and average time required for completion.

## Sample size

We will recruit up to 30 patients from QEHB for this study based on the recommendations found in literature.[25] The process of improving the usability of any system is an iterative one[22]; therefore, a minimum of two testing cycles will be conducted with the patients. The findings from the first test cycle will guide the process of improving the ePROM system before the second cycle is conducted.

## Study 2: qualitative study

This study will explore the views of stakeholders on the optimal use and administration of the CKD ePROM system.

### Study 2a: patient interviews
#### Study objectives
► To determine which symptoms patients with CKD find most bothersome.
► To explore how acceptable ePROMs are.
► To determine how often patients will be willing to complete the ePROM and their preferred method of completion, that is, PC, smartphone, tablet, telephone voice recognition or paper completion.
► To explore the likely factors that may improve or discourage the completion of ePROMs.
► To explore how they would like to receive feedback from the clinical team regarding the ePROMs they provide.

#### Data collection
Semi-structured face-to-face interviews will be arranged to either coincide with patients' scheduled clinic visit or held on a separate day if preferred. The option of a telephone interview will also be given.

A topic guide will be used to provide a general direction for each interview and ensure that important issues are covered while allowing enough flexibility to capture other relevant themes that may arise during any session.

Interviews will be recorded using an encrypted digital audio recorder and transcribed by a professional transcription company.

#### Data analysis
The transcripts will be analysed by the CI using the Nvivo 10 software package by QSR International. Thematic analysis of the data will be conducted following the six steps described by Braun and Clarke.[26] The process of analysis will begin with the CI 'actively' reading and engaging with the data set (ie, searching for patterns and meanings). The next phase will be the initial coding of the raw transcript data using the QSR software. Extracts will be coded

inclusively (ie, a little surrounding data will be kept to retain contextual meaning).[27]

Phase 3 will involve the analysis and organisation of codes into potential themes. These initial themes will be revised and refined in the fourth phase on two levels. The first is at the level of coded data extracts to ensure they are coherent for each theme. The second level of analysis is to ensure the themes reflect the data set. During this phase, redundant codes and themes may be removed, revised or merged as required. Phase 5 will involve the definition of what each theme is and what it is not, and its importance in relation to the entire data and the research questions. The themes will be considered individually as well as in relation to other themes to ensure overlaps are kept to a minimum. The final phase will be the production of the study report.

The project team (Independent of the CI) will randomly review a sample of transcripts for verification purposes.

Data analysis will be carried out simultaneously with data collection, and both will continue until no new themes emerge from the further analysis, that is, data saturation has been reached.[28]

Respondent validation will be undertaken, whereby a summary of the main points arising from the interview will be sent to each participant for comments.

#### Sample size
Based on experience from previous similar qualitative studies conducted by the research team, recruitment will continue until a target sample size of between 15 and 30 patient participants is attained or until saturation is achieved.

### Study 2b: focus groups with clinicians and other professional staff
Focus group discussions will be held with clinicians who manage patients with CKD at the Renal Unit, QEHB, members of the myHealth team and hospital management staff as required.

#### Study objectives
► To evaluate and rate the relevance of the items of the ePROM questionnaires with clinical staff (content validation).
► To determine those factors that may improve or discourage the use of ePROM data by clinicians.
► To determine clinicians' preferred method of displaying ePROM data.

#### Data collection
An independent member of staff will serve as chief moderator and direct the discussions while the CI will act as assistant moderator, making notes and observing the interactions within the groups. The discussions will be allowed to develop with minimum interference following a topic guide to ensure that all the main points are covered. Focus group sessions will be recorded using an encrypted digital audio recorder and transcribed by a professional transcription company.

Face-to-face or telephone interviews with clinicians and other professional staff may be conducted to explore their views on the use of PROMs. These interviews will require a maximum of 1 hour.

### Sample size
We will aim to include up to 15 participants in up to two focus groups (seven to eight participants in each group) and if necessary interview the same number of participants.

### Data analysis
Thematic analysis will be conducted (see analysis of patient interviews for details).

## Study 3: ePROM validation
The purpose of this study is to evaluate the measurement properties of the electronic versions of the KDQOL-36 and the IPOS-Renal against the EQ-5D-5L and clinical data. At the end of this study, the most suitable questionnaire(s) would be taken forward for formal feasibility testing.

### Study objectives
- To determine the reliability and validity of the ePROM questionnaires.
- To determine the ability of the ePROM questionnaires to detect change in a patient's health over a period of time.
- To determine which of the two questionnaires is most suitable to take forward for formal feasibility testing.

### Data collection
Patients will be registered on 'myhealth@QEHB' in order to access the ePROM system. There will be the option of completing paper versions if preferred. Time for questionnaire completion may be influenced by patients' symptoms but should require no more than 1 hour.

All participants will be asked to complete the two questionnaires three times: at study entry, at 2 weeks after initial completion and at 6 months after initial completion. Completing the questionnaires at these time points will facilitate the comprehensive assessment of psychometric properties.[8] For definitions of measurement properties, see table 2.

### Data analysis
Quantitative data will be analysed using statistical software such as STATA. Where appropriate, analysis will be conducted separately for patients with CKD stages 4 and 5 and patients on dialysis.

A disclaimer statement will be included in the patient information sheets for the validation study informing the patients that the questionnaires will not be assessed until the end of the study; therefore, patients should inform their clinician (eg, general practice or renal services) of any healthcare needs for management.

## Psychometric evaluation
Classical test theory (CTT) and the Rasch measurement model[29] will be used to evaluate the psychometric properties of the ePROM versions of the KDQOL-36 and IPOS-Renal. CTT is a traditional approach to questionnaire development, which postulates that a person's observed score consists of their true score plus an additional measurement error score, and the underlying assumption is that this relationship is additive.[30 31] This additive model involves the summation of item ratings on a Likert-type scale to obtain a total score; however, the values of the true score and error score cannot be determined and CTT does not describe the hierarchy of the items.[30–32]

Rasch analysis[29] is one method of evaluating measurement tools to ensure they deliver reliable and valid measurement and is increasingly being used in clinical research and practice for refinement and development of PROMs.[33] The Rasch model operationalises axioms of additive conjoint measurement and tests the extent to which PROMs are unidimensional.[34] Fit to the Rasch model is established through a number of fit statistics.[35 36] Analysis is an iterative process identifying and studying anomalies in the data and the extent to which data conforms to the Rasch model. The degree of fit achieved will identify the extent to which KDQOL-36 and IPOS-Renal demonstrate construct validity, unidimensionality and reliability.[37] When data fit the Rasch model, it confirms that the PROMs are unidimensional and summation of scores from the KDQOL-36 and IPOS-Renal is legitimate.[35]

CTT will be used to evaluate the reliability, construct validity and responsiveness while Rasch analysis will be done to complement the CTT assessment of structural validity of the two questionnaires.

## CTT methods
### Factor analysis
#### Structural validity
Exploratory factor analysis will be used to evaluate the factor structure of the KDQOL-36 and the IPOS-Renal.[38 39] This will be conducted using principal component analysis (PCA) with orthogonal Varimax rotation of quadrants.[38] Factors will be identified based on the Scree test and the percentage of variance accounted for by a particular factor.[40 41] Eigenvalues measure the amount of variation and factors will be required to have a minimum eigenvalue of 1.0.[40] Subsequently, a confirmatory factor analysis will be conducted to test whether the hypothesised factor models of the questionnaires are supported by actual data.[38 42]

### Reliability
#### Internal consistency
Using baseline scores, Cronbach's alpha[43] will be calculated for the total scale and subscale scores of the KDQOL-36 and IPOS-Renal. Alpha values 0.70–0.90 will be deemed acceptable[44–46] An 'if item deleted' analysis

**Table 2**  Definitions of domains, measurement properties and aspects of measurement properties

| Domain | Measurement property | Aspect of measurement property | Definition |
|---|---|---|---|
| Reliability | Reliability (extended definition) | | The degree to which the measurement is free from measurement error |
| | | | The extent to which scores for patients who have not changed are the same for repeated measurement under several conditions: for example, using different sets of items from the same HR-PROs (internal consistency), over time (test–retest) by different persons on the same occasion (inter-rater) or by the same persons (ie, raters or responders) on different occasions (intrarater) |
| | Internal consistency | | The degree of the interrelatedness among the items |
| | Reliability | | The proportion of the total variance in the measurements which is because of 'true'* differences among patients |
| | Measurement error | | The systematic and random error of a patient's score that is not attributed to true changes in the construct to be measured |
| Validity | | | The degree to which an HR-PRO instrument measures the construct(s) it purports to measure |
| | Content validity | | The degree to which the content of an HR-PRO instrument is an adequate reflection of the construct to be measured |
| | | Face validity | The degree to which (the items of) an HR-PRO instrument indeed looks as though they are an adequate reflection of the construct to be measured |
| | Construct validity | | The degree to which the scores of an HR-PRO instrument are consistent with hypotheses (for instance with regard to internal relationships, relationships to scores of other instruments or differences between relevant groups) based on the assumption that the HR-PRO instrument validly measures the construct to be measured |
| | | Structural validity | The degree to which the scores of an HR-PRO instrument are an adequate reflection of the dimensionality of the construct to be measured |
| | | Hypotheses testing | Idem construct validity |
| | | Cross-cultural validity | The degree to which the performance of the items on a translated or culturally adapted HR-PRO instrument are an adequate reflection of the performance of the items of the original version of the HR-PRO instrument |
| | Criterion validity | | The degree to which the scores of an HR-PRO instrument are an adequate reflection of a 'gold standard' |
| Responsiveness | | | The ability of an HR-PRO instrument to detect change over time in the construct to be measured |
| | Responsiveness | | Idem responsiveness |
| Interpretability† | Interpretability† | | The degree to which one can assign qualitative meaning—that is, clinical or commonly understood connotations—to an instrument's quantitative scores or change in scores |

Reproduced with permission from Caroline Terwee, COSMIN.
*The word 'true' must be seen in the context of the CTT, which states that any observation is composed of two components—a true score and error associated with the observation. 'True' is the average score that would be obtained if the scale were given an infinite number of times. It refers only to the consistency of the score and not to its accuracy.[12]
†Interpretability is not considered a measurement property but an important characteristic of a measurement instrument.
CTT, classical test theory; HR-PRO, health-related patient-reported outcome.

will be conducted to identify whether any items should be dropped from the scale.[47]

### Test–retest reliability

The completion of the electronic questionnaires 2 weeks after the initial completion will allow assessment of the stability of the questionnaires.[48] Intraclass correlation coefficients (ICCs) for agreement will be calculated on subscale and total scores using a two-way random effects model.[49 50] ICC values >0.75 will indicate excellent test–retest reliability, values 0.40–0.75 will be considered good, while values <0.4 will indicate weak agreement.[30 48 51]

### Measurement error

As we are not aware of values for measurement error and minimally important clinical change (MIC) for our population of interest, these will be calculated in this study. Measurement error will be calculated using the SE of measurement.[49 52] The MIC will be determined for patients who commence dialysis within the study period using a patient-reported anchor-based method. This is regarded as the ideal method for calculating the MID as it captures the patients' values directly.[53] We will compare changes in measurement scores with a patient-reported global rating of change scale as our reference 'anchor'.[53–55] The measurement error and the MIC will assist with the assessment and interpretation of treatment outcomes and effects.[56]

### Construct validity
### Convergent validity (hypothesis testing)

We have formulated the following hypotheses to test in order to establish convergent validity. Pearson's or Spearman's correlation coefficients will be calculated for correlations as appropriate. Pearson's or Spearman's correlation coefficients >0.40 will be considered acceptable for scales that are theoretically related.[57 58]

### Hypothesis 1

Each item of the KDQOL-36 and the IPOS-Renal will have a positive correlation ≥0.40 with its own hypothesised subscale after correction for overlap.[59] IPOS-Renal items will be correlated with their subscales once these have been established by factor analysis.

### Hypothesis 2

The generic (SF-12) and the disease-specific domains of the KDQOL-36 will have weak to moderate positive correlations with each other as they are designed to assess different aspects of health-related quality of life.[60 61]

### Hypothesis 3

Each subscale score of the KDQOL-36 will have positive correlations with the overall health rating score (question 1 of KDQOL-36).[62]

### Hypothesis 4

The generic (SF-12) subscales, the physical component summary and the mental component summary of the KDQOL-36 will have higher positive correlations with the utility scores of the EQ-5D-5L than the kidney-specific subscales of the KDQOL-36 and the symptom-specific scales of the IPOS-Renal.

### Hypothesis 5

Clinical parameters specific to kidney disease such as the eGFR will correlate better with dialysis-targeted dimensions of the KDQOL-36 and the IPOS-Renal than with generic dimensions of KDQOL-36.[63]

### Hypothesis 6

The utility scores of the EuroQol Visual Analogue Scale will have a high positive correlation with the overall health rating scores of the KDQOL-36.

### Hypothesis 7

The comparisons of the means of the lowest scoring 25th percentile and the higher scoring 75th percentile for each disease-specific subscale of the KDQOL-36 will be statistically significant (p values <0.05, using the Mann-Whitney U test).[64]

### Responsiveness

The questionnaires will be administered 6 months after the initial completion in order to assess the ability of the questionnaires in detecting changes in patients' condition. Using Pearson's correlation coefficient, we will test three hypotheses for responsiveness based on changes in scores as recommended by the Consensus-based Standards for the selection of health Measurement INstruments (COSMIN) group.[65]

### Hypothesis 1

There will be significant changes in the QOL scores of patients who switch from conservative management to RRT within this period. Therefore, the QOL scores before and after commencing RRT will be compared using the Wilcoxon signed-rank test.[66]

### Hypothesis 2

Changes in KDQOL-36 scores for patients who switch to RRT from conservative care will correlate negatively with changes in their creatinine values and correlate positively with changes in residual renal function and serum albumin.[63 65]

### Hypothesis 3

There will be positive correlations between the global rating scale and the changes in KDQOL-36 scores for patients who switch to RRT from conservative care.[65]

In addition to these hypotheses, effect sizes (ES) and standardised response mean (SRM) will be calculated for patients with CKD stages 4 and 5 who were initially managed conservatively but progressed to renal failure (on RRT) during the study period.[67 68]

Higher ES or SRM indicate greater responsiveness and values up to 0.2 will be regarded as small, 0.5 moderate and 0.8 as substantial according to Cohen's criteria.[69] Receiver

operating characteristic curves will be used to establish a cut point for predicting transition to RRT.

## Application of Rasch analysis

The underlying assumption with the Rasch model is that individual items capture a single underlying trait, and therefore the summation of items from the KDQOL-36 or IPOS-Renal forms unidimensional scales. Rasch analysis is an iterative process that identifies and studies anomalies in the data and the extent to which KDQOL-36 or IPOS-Renal data conform to the Rasch model and therefore the extent to which the instrument is unidimensional. Fit will be established using a variety of indicators and fit statistics.[70] The Rasch Unidimensional Measurement Model software (RUMM2030)[71] will be used to analyse KDQOL-36 and IPOS-Renal data.

KDQOL-36 has a mixture of dichotomous and polytomous responses whereas the IPOS-Renal items are all polytomous using a Likert response format (see table 1). Affirmation of response categories by respondents should follow a logical sequence. As their perceived level of health improves/deteriorates, then responses should reflect this by affirming higher or lower scoring categories, moving from a score of 1, to 2, then 3, etc, on any item.[72] Rasch-Andrich thresholds are the points between adjoining categories where the probability of affirming either category is 50/50, when responders' perceived level of health is equidistantly captured by adjoining categories.[73] Where there is agreement with this expected response hierarchy, thresholds appear ordered, and disordered thresholds are observable as a lack of consistency.[73] Disordered thresholds can suggest poorly defined or redundant scoring categories, and therefore conceptual distinctions between categories may be imprecise. Responders then find it difficult to assign a category to their perceived health status or QOL.

Targeting will be established by examining the extent to which distributions of participants perceived QOL/health status and levels of health identified by KDQOL-36 and IPOS-Renal items are analogous. The position of each responder and item on the underlying construct is defined as the person's ability and the item's difficulty.[31] Therefore, responders with low levels of perceived health or QOL should only affirm items and scoring categories which capture low levels of health. The item which captures the average level of ability will be identified as having zero logits by RUMM2030. Therefore, when person and items are appropriately targeted, person mean location scores should approximate the zero value of the item locations. A positive mean value for responders' estimated locations will suggest that responders' average levels of health are higher than the average on the KDQOL-36. Conversely, negative person locations will confirm the opposite to be true.[35]

The person separation index (PSI) uses the logit values to estimate the internal reliability of the KDQOL-36 and IPOS-Renal and is conceptually equivalent to Cronbach's alpha. It identifies the extent to which the instrument is able to discriminate between groups with different health states and the precision of the estimate for each person.[74] Minimum PSI value suggested for group use is 0.70 and for individual use 0.85.[35]

Individual tests of fit for each person and item will reflect the difference between responders observed and expected responses if data fit the Rasch model. RUMM2030 will automatically cluster responders into equivalent size groups (class intervals) according to their overall ability. A number of statistics use these class intervals including $\chi^2$ statistics and residual values.[75] Residuals are summations of individual item or person deviations from expected fit to the Rasch model, standardised as a z-score. Residual scores between ±2.5 indicate adequate fit to the Rasch model.[75]

A $\chi^2$ statistic will compare this difference, with a summed $\chi^2$ for each class interval contributing to the overall $\chi^2$ for that item. The $\chi^2$ for all items will then be summed to demonstrate the overall item-interaction statistic. A non-significant $\chi^2$ interaction statistic will indicate theoretical fit to the Rasch model.[75] A significant $\chi^2$ will indicate the need for further evaluation to establish potential causes of misfit.

Differential item functioning (DIF) is a form of item bias that can affect fit to the model. DIF manifests itself as responses to individual items by sample subgroups (eg, gender or age group) being inconsistent with their overall perceived level of health.[76] DIF will be identified using ANOVA and statistically significant probability (p<0.05, or a Bonferroni-corrected level). DIF for gender, age group and kidney disease stage will be examined.

Item independence is an underlying principle of the Rasch model.[37] Response dependency occurs when a person's response to one item determines the response to another item, and therefore responses are not independent of each other.[35] Residual correlation metrics (<0.3) will identify if response dependency is an issue.

Once the 'Rasch factor' is extracted, leftover residuals should not contain any patterns in the data.[35] A PCA of the residuals will detect if multi-dimensionality is an issue.[37] Subsets of items identified by the negative and positive correlations from the PCA will be used to compare estimates of responders' health states on the two subsets. If no significant difference in the estimates is identified using independent t-test, then unidimensionality is assured. Tennant and Conaghan[35] also state that the percentage of tests outside the range of −1.96 to 1.96 should not exceed 5%. Confidence intervals for the binomial test of proportions will be used; this is based on the number of significant tests and the lower bound should overlap the 5% expected value for a non-significant test in order to confirm unidimensionality.[35] If responders' health state estimates are found to be significantly different in more than 5% of cases, this will indicate that the subtests are measuring different but related aspects of health states.[35] Where the scale is being used to measure changes over time, then using different but related subscales might be more appropriate.[75]

Finally, if the data fit the model, patient and item parameter estimates will be positioned on the same log-odds units (logits) scale, although as independent parameters allowing a linear transformation of the raw scores to be used.[75] Therefore, estimates of a patient's level of health can be derived from the KDQOL-36 and IPOS-Renal data with confidence.

### Sample size
There are various schools of thought regarding sample size requirements for validation studies.[77–80] We aim to recruit at least 180 patients based on the recommendations found in contemporary literature outlining psychometric best practice (5 to 10 times the number of variables in any given multivariate statistical model).[77 78]

### Reimbursement and withdrawal
*Reimbursement*
All patients will be reimbursed for their time with a £20 gift card.[81 82] Light refreshments will be provided for focus group participants.

*Withdrawal*
Patient participants will be informed they have the right to freely withdraw from the study, for any reason, at any time prior to their data being integrated into the data set. They will not be required to supply a reason for their withdrawal and the decision will have no effect on their future medical care.

As focus group participants (clinicians and other professional staff) will be audio recorded as a group, they will be informed it will be impossible to withdraw their data during or after a focus group discussion. They will not be required to supply a reason for their withdrawal before a focus group discussion and the decision will have no effect on their employment.

### DISCUSSION
PROMs can be completed electronically, making it possible to remotely monitor symptoms in patients with CKD and generate 'real time' data, which may assist clinicians with the tailoring of treatment to the needs of each patient.[12] The use of ePROMs could potentially foster patient–clinician communication and further support shared-decision between patients with CKD and their clinicians.[13 14] The regular completion of ePROMs may decrease the need for stable patients with acceptable ePROM data to attend clinical appointments, thus sparing them the financial burden and physical stress of travelling. This might free up appointment times for patients who actually need to be seen in clinic.[12] In this manner, the use of PROMs may significantly improve the quality of life of patients with CKD.

At each stage of the project, when necessary, PAG meetings will be convened for their input and drafts of publication manuscripts reviewed by members of the group before submission to journals.

By employing a mixed-methods approach, the PRO-trACK project will provide evidence of the feasibility and validity of the ePROM system in patients with advanced CKD.

### ETHICS AND DISSEMINATION
#### Ethics and data management
This project was approved by the West Midlands Edgbaston Research Ethics Committee (Reference 17/WM/0010) and received Health Research Authority (HRA) Approval on 24 February 2017. It has also been included in the National Institute for Health Research (NIHR) Clinical Research Network (CRN) Portfolio (ID 33117).

Participant data (whether in electronic or paper format) will be acquired, anonymised, transferred and stored according to the Data Protection Act 1998[83]; the Confidentiality—NHS Code of Practice[84]; the Caldicott principles[85]; the University of Birmingham Code of practice for research[86]; and the University of Birmingham Guidance on Out of Hours Activities and Lone Working.[87]

Only members of the research team will have access to the project data. The exception will be permissions given to authorised regulatory personnel in order to conduct audits and inspections on behalf of the ethics committee.

#### Dissemination
The findings of the project will be provided to the Informatics Team and the Nephrology Unit at the QEHB as required. Participants will be given a summary of the findings, with reference to the full reports if desired.

Research article(s) based on the findings of the studies will be written and submitted for publication to peer-reviewed journals, and all contributors and their contributions to the study will be acknowledged. We will also disseminate our findings at seminars and conferences both nationally and internationally.

**Author affiliations**
[1]Centre for Patient-Reported Outcomes Research, University of Birmingham, Birmingham, UK
[2]Institute of Applied Health Research, University of Birmingham, Birmingham, UK
[3]Department of Renal Medicine, University Hospitals Birmingham NHS Foundation Trust, Queen Elizabeth Hospital Birmingham, Birmingham, UK
[4]Patient Advisory Group Member, Centre for Patient-Reported Outcomes Research, University of Birmingham, Birmingham, UK

**Acknowledgements** The authors thank all the members of the Patient Advisory Group (PAG), Centre for Patient-Reported Outcomes Research (CPROR), University of Birmingham, for their comments and suggestions during the development of this project.

**Contributors** MC is the guarantor for this project. The project was conceived by MC, DK, PC and TM and designed by MC, DK, PC, TM, OLA, MD and AS. OLA drafted the protocol manuscript. The manuscript was reviewed by MC, DK, PC, TM, OLA, MD, AS, NM, GP, RV, JW and KS. The final draft was approved by all authors.

**Funding** This project is funded as part of the Health Foundation's PhD Awards for Improvement Science. The Health Foundation is an independent charity working to improve the quality of healthcare in the UK. The Health Foundation was not involved in any other aspect of the project. TM is partly funded by the National Institute for Health Research (NIHR) through the Collaborations for Leadership in Applied Health Research and Care for West Midlands (CLAHRC/WM). This paper presents independent research and the views expressed in this publication are not necessarily those of the NIHR, the Department of Health, NHS Partner Trusts, University of Birmingham or the CLAHRC/WM Management Group.

**Competing interests** None declared.

**Ethics approval** West Midlands Edgbaston Research Ethics Committee (Reference 17/WM/0010).

**Provenance and peer review** Not commissioned; externally peer reviewed.

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
