## [Reviewer comments · BMJ Open]

ARTICLE DETAILS

TITLE (PROVISIONAL)	Using Patient-Reported Outcome measures (PROMs) to promote quality of care and safety in the management of patients with Advanced Chronic Kidney Disease (PRO-trACK Project) – A mixed-methods project protocol
AUTHORS	Aiyegbusi, Olalekan; Kyte, Derek; Cockwell, Paul; Marshall, Tom; Dutton, Mary; Slade, Anita; Marklew, Neil; Price, Gary; Verdi, Rav; Waters, Judi; Sharpe, Keeley; Calvert, Melanie

VERSION 1 - REVIEW

REVIEWER	Elizabeth Gibbons Health Services Research Unit, Nuffield Department of Population Health University of Oxford
REVIEW RETURNED	15-Mar-2017

GENERAL COMMENTS	This is a comprehensive mixed methods study and the protocol illustrates the different studies within it. There seems to be overlap between the usability testing study and the qualitative interviews with patients and it is not clear if the patients in the latter will be given access to the ePROM system or whether their general views are being sought. Also- no details are provided about the selection of PROMs being used in the ePROM and evidence to support the choice of them. Details should be provided of the PROMs and and licensing issues stated. The interviews with staff will explore the content of the PROMs to be included - it is not clear if this will be done prior to the development of the ePROM system. Psychometric analysis seems according to recommendations. It is not clear how responsiveness will be evaluated, whether this is correlation between changes in clinical parameters or patients perceived change- using a patient reported transition item. A clearer understanding of how the final PROM selection will be addressed would be helpful. Overall, an interesting study addressing current PROMs research agenda
---

REVIEWER	Angela M. Stover, PhD University of North Carolina at Chapel Hill, USA
REVIEW RETURNED	06-Apr-2017

GENERAL COMMENTS	The authors are submitting a mixed methods protocol related to ePROMs in chronic kidney disease. It is great to see that members of a patient advisory group are part of the project and co-authors on the protocol.
--

Major:

The protocol is evaluating 3 questionnaires for potential use in chronic kidney disease. However, the names of the questionnaires, number of items on each questionnaire (and number of subscales), and number of response option categories do not appear to be stated. Are these generic QOL measures or specific to CKD? What construct(s) are they assessing specifically?

The given sample size for Rasch modeling (n=180) cannot be evaluated for adequacy without knowing the total number of items, subscales, and number of response options. It would be helpful to add subject numbers for all aims to the abstract too.

What proportion of patients at the Univ. Hospitals Birmingham NHS are signed up for the patient portal and are these patients representative of chronic kidney disease (particularly relevant for Aim 3). In the U.S., patient portal enrollment is only about 30-35% and users tend to be affluent with high education levels. Will you be doing any purposeful sampling to ensure you are assessing individuals with lower education or literacy?

- On page 17, protocol states that purposeful sampling will be done for ethnic minorities. Specify what percentage you are aiming for.
- What about other important demographic variables like age? There may be differences in perceptions of acceptability and value of ePROMs for middle aged vs. older CKD patients.

It seems like the protocol could be tightened up. For instance, there is a lot of repetition between overview, research objectives, and methods sections.

The references for study 3 appear to be inadequate for both the classical test theory and Rasch analyses. For instance, on page 20, the FDA guidance document is cited to support your time points of study entry, 2 weeks, and 6 months. Measurement papers themselves would be better.

On page 21, the sections on measurement error, convergent validity, responsiveness, and Rasch are confusing.

- In measurement error section, it is common practice to specify and cite the MIC and SDC you will be using. Are these established for your 3 questionnaires?

- In convergent validity, it says pearson correlations will be used. Pearson correlations between the 3 scales you are interested in? Is there a gold standard 4th questionnaire you will use to determine if the first 3 questionnaires are measuring similar concepts? How will you deal with subscales that may be measuring different concepts or symptoms? Is there any literature on convergent and divergent validity for your 3 scales already available? What cut-off will you use to determine how "high" do correlations need to be to provide evidence for convergence? Consider adding your cut-off points to Table 1 to make it easy for the reader.

- How will responsiveness be assessed? Protocol on page 21 states that questionnaires will be at 6 months but this alone would not be sufficient to look at responsiveness. "Clinical and laboratory data" are vaguely mentioned for comparisons. More detail is warranted here to determine if they are appropriate.

- For Rasch, protocol states that the analysis will examine "suitability" of items and to identify redundant items. Suitability seems like a strange way to describe Rasch; it makes it sound like

	acceptability to patients. Without knowing how many items there are on the questionnaires and the number of subscales (and response option levels), it cannot be determined if 180 people is sufficient. Additionally, no standard Rasch references are cited. Sample size is justified by the COSMIN protocol for systematic reviews (ref #27), which seems very odd. More detail about the Rasch analyses is warranted. Minor: Abstract: spell out PRO-trACK acronym On page 15, it states that patients will be “randomly enrolled into groups of 3-5 patients per group.” Consider a different word choice for “randomly,” which suggests randomization. On page 20, there is a disclaimer statement for patients that results will not be fed back to clinicians for Aim 3. Isn't this true for aims 1-2 too?
--	--

REVIEWER	Helen Noble Queens University Belfast, UK
REVIEW RETURNED	06-Apr-2017

GENERAL COMMENTS	Thank you for submitting this interesting protocol. If the BMJ Open accepts protocols I suggest it is relevant and well written and would be of value to readers. It may be more relevant for a journal such as BMC nephrology who do publish protocols. Some small comments as I presume that this study has already been peer reviewed: Abstract and main text: I am not sure why you call this a project made up of 3 studies. It appears to be a research study made up of 3 parts. I suggest under dissemination you mention international dissemination pg 4 - strengths and limitations Why is the ePROM only intended for stage 4 & 5 CKDS patients? Top of pg 8 - one study in 3 parts? Pg 11- line 49. Something brief about the renal risk calculator would be of interest to readers. Line 54 - requires reference. I haven't come across the statement that ESRD is when eGFR falls below 10. pg 12 - line 5 - be more tentative.....they don't always have the highest symptom burden pg 329 - how much funding is supporting the study?
--

VERSION 1 – AUTHOR RESPONSE

Reviewer 1 – Dr Elizabeth Gibbons

We are grateful for the comments of Dr Gibbons and have addressed them below

1. There seems to be overlap between the usability testing study and the qualitative interviews with patients and it is not clear if the patients in the latter will be given access to the ePROM system or whether their general views are being sought.

Response: Whilst the usability testing and qualitative interviews are related, they are distinct studies;

the usability testing will focus on the actual experience of patients as they test the ePROM system while the qualitative interviews will explore the broader opinions of patients on the use of ePROMs in the NHS. (Paragraph 4, page 6)

Concerning patient access, we have now included a paragraph in the manuscript. 'We will aim to recruit different sets of eligible patients for each study in order to minimize participant burden. However, if patients voluntarily express an interest, they will be allowed to participate in more than one study if the clinical team is satisfied with their health status.' (Paragraph 4, page 11)

2. Also- no details are provided about the selection of PROMs being used in the ePROM and evidence to support the choice of them. Details should be provided of the PROMs and and licensing issues stated.

Response: Selection of measures was informed by i) a systematic review of measurement properties of PROMs used in CKD patients and ii) feedback from the PAG members. The systematic review found evidence to support the use of the KDQOL-SF and KDQOL-36. However, these two measures were validated by very few studies in our population of interest (stages 4 and 5 CKD). The review also identified the IPOS-Renal, which is currently undergoing validation through use in a number of renal units in the UK. The Patient Advisory Group (PAG) members chose the KDQOL-36 and IPOS-Renal. Therefore, the decision was made to validate these in the pre-dialysis population (stage 4 and 5). (Page 5, paragraph 3)

The KDQOL-36 and the IPOS-Renal are free to use without charge as long as the developers are appropriately acknowledged and cited. The EQ-5D requires prior written consent and payment of licensing fees (if applicable). A license will be obtained for this project. (Page 6, paragraph 2)

3. The interviews with staff will explore the content of the PROMs to be included - it is not clear if this will be done prior to the development of the ePROM system.

Response: We thank the reviewer for this comment and this has been clarified.

'The content validation of the KDQOL-36 and the IPOS-Renal by patients and clinicians during the qualitative study as well as the results of the validation study will assist the research team with the final decision on which measure to take forward for the final version of the ePROM system.' (Page 6, paragraph 4)

4. Psychometric analysis seems according to recommendations. It is not clear how responsiveness will be evaluated, whether this is correlation between changes in clinical parameters or patients' perceived change- using a patient reported transition item.

Response: We have now clarified the manuscript. Responsiveness will be assessed using both methods. (Pages 28 & 29)

5. A clearer understanding of how the final PROM selection will be addressed would be helpful.

Response: We have revised to include the following statement.

'The content validation of the KDQOL-36 and the IPOS-Renal by patients and clinicians during the qualitative study as well as the results of the validation study will assist the research team with the final decision on which measure to take forward for the final version of the ePROM system.' (Page 6, paragraph 4)

Reviewer: 2 – Angela M. Stover, PhD

We are grateful for the comments of Dr Stover and have addressed them below

1. The protocol is evaluating 3 questionnaires for potential use in chronic kidney disease. However, the names of the questionnaires, number of items on each questionnaire (and number of subscales), and number of response option categories do not appear to be stated. Are these generic QOL measures or specific to CKD? What construct(s) are they assessing specifically?

Response: We have revised to now include a table providing this information. (Page 8, Table 1)

2. The given sample size for Rasch modeling (n=180) cannot be evaluated for adequacy without knowing the total number of items, subscales, and number of response options. It would be helpful to add subject numbers for all aims to the abstract too.

Response: Please see Page 8, Table 1. Subject numbers have been added to the abstract on Page 2.

3. What proportion of patients at the Univ. Hospitals Birmingham NHS are signed up for the patient portal and are these patients representative of chronic kidney disease (particularly relevant for Aim 3). In the U.S., patient portal enrollment is only about 30-35% and users tend to be affluent with high education levels. Will you be doing any purposeful sampling to ensure you are assessing individuals with lower education or literacy?

Response: Around 1200 renal patients are currently signed up. At the moment this is a voluntary system. Although our previous work does not show an influence of socio-economic status (SES) on outcomes for CKD (Jesky 2016), we will be mindful of sample diversity in relation to socio-demographic variables such as age, gender, ethnicity and other relevant socio-demographic factors. We will collect data on participant characteristics to monitor this as recruitment and qualitative data collection progress. Although we appreciate that patient populations and research samples do not always represent such diversity, we will try to employ recruitment strategies that optimise our ability to recruit a diverse patient sample. (Paragraph 4, page 11)

4. On page 17, protocol states that purposeful sampling will be done for ethnic minorities. Specify what percentage you are aiming for.

Response: '.....we will recruit up to 30% of the study participants from minority ethnic groups....'. (Page 11, paragraph 4)

5. What about other important demographic variables like age? There may be differences in perceptions of acceptability and value of ePROMs for middle aged vs. older CKD patients.

Response: Please see comment 3.

6. It seems like the protocol could be tightened up. For instance, there is a lot of repetition between overview, research objectives, and methods sections.

Response: We have removed the overview sections and reformatted the other sections. (Pages 12 – 32)

7. The references for study 3 appear to be inadequate for both the classical test theory and Rasch analyses. For instance, on page 20, the FDA guidance document is cited to support your time points of study entry, 2 weeks, and 6 months. Measurement papers themselves would be better.

Response: We thank the reviewer for this comment. More details have been provided. (Pages 23 & 32)

8. On page 21, the sections on measurement error, convergent validity, responsiveness, and Rasch are confusing.

In measurement error section, it is common practice to specify and cite the MIC and SDC you will be using. Are these established for your 3 questionnaires?

Response: The sections have been clarified. (Pages 25 & 31) As we are not aware of the availability of these MIC and SDC values for our population of interest, we will be calculating them.

9. In convergent validity, it says Pearson correlations will be used. Pearson correlations between the 3 scales you are interested in? Is there a gold standard 4th questionnaire you will use to determine if the first 3 questionnaires are measuring similar concepts?

Response: This has now been clarified. 'The PAG members chose the KDQOL-36 and IPOS-Renal. Therefore, the decision was made to validate the electronic versions of these in the pre-dialysis population (stage 4 and 5). The EuroQol 5-dimension 5-Level (EQ-5D-5L) questionnaire will be used as a comparison measure for this validation study.' (Page 6, paragraph 1)

The purpose of this study is to evaluate the measurement properties of the electronic versions of the KDQOL-36 and the IPOS-Renal against the EQ-5D-5L and clinical data. At the end of this study, the most suitable questionnaire(s) would be taken forward for formal feasibility testing. (Page 21, paragraph 1)

10. How will you deal with subscales that may be measuring different concepts or symptoms? Is there any literature on convergent and divergent validity for your 3 scales already available?

Response: As the validation study will be using already developed questionnaires we do not have any influence on the constructs chosen by the developers. Significant and relevant correlations between constructs will be reported as previous validation studies have done.

Selection of measures was informed by i) a systematic review of measurement properties of PROMs used in CKD patients and ii) feedback from the PAG members. The systematic review found evidence to support the use of the KDQOL-SF and KDQOL-36. However, these two measures were validated by very few studies in our population of interest (stages 4 and 5 CKD). The review also identified the IPOS-Renal, which is currently undergoing validation through use in a number of renal units in the UK. The Patient Advisory Group (PAG) members chose the KDQOL-36 and IPOS-Renal. Therefore, the decision was made to validate these in the pre-dialysis population (stage 4 and 5). (Page 5, paragraph 3)

11. What cut-off will you use to determine how "high" do correlations need to be to provide evidence for convergence? Consider adding your cut-off points to Table 1 to make it easy for the reader.

Response: We have provided this information in text for all the relevant calculations and not in Table 1 as suggested as our permission to reproduce the table does not cover modification of its contents. (Pages 22 – 32)

12. How will responsiveness be assessed? Protocol on page 21 states that questionnaires will be at 6 months but this alone would not be sufficient to look at responsiveness. "Clinical and laboratory data" are vaguely mentioned for comparisons. More detail is warranted here to determine if they are appropriate.

Response: These details have been provided.

The questionnaires will be administered 6 months after the initial completion in order to assess the ability of the questionnaires in detecting changes in patients' condition. Using Pearson's correlation coefficient, we will test three hypotheses for responsiveness based on changes in scores as recommended by the COSMIN group.

Hypothesis 1 – There will be significant changes in the QOL scores of patients who switch from conservative management to RRT within this period. Therefore, the QOL scores before and after commencing RRT will be compared using the Wilcoxon signed-rank test.

Hypothesis 2 – Changes in KDQOL-36 scores for patients who switch to RRT from conservative care will correlate negatively with changes in their creatinine values and correlate positively with changes in residual renal function and serum albumin.

Hypothesis 3 – There will be positive correlations between the global rating scale and the changes in KDQOL-36 scores for patients who switch to RRT from conservative care.

In addition to these hypotheses, effect sizes (ES) and standardized response mean (SRM) will be calculated for patients with CKD stages 4 & 5 who were initially managed conservatively but progressed to renal failure (on RRT) during the study period.

Higher ES or SRM indicate greater responsiveness and values up to 0.2 will be regarded as small; 0.5 moderate and 0.8 as substantial according to Cohen's criteria.⁷⁴ Receiver operating characteristic (ROC) curves will be used to establish a cut point for predicting transition to RRT". (Pages 27 & 28)

13. For Rasch, protocol states that the analysis will examine "suitability" of items and to identify redundant items. Suitability seems like a strange way to describe Rasch; it makes it sound like acceptability to patients. Without knowing how many items there are on the questionnaires and the number of subscales (and response option levels), it cannot be determined if 180 people is sufficient. Additionally, no standard Rasch references are cited. Sample size is justified by the COSMIN protocol for systematic reviews (ref #27), which seems very odd. More detail about the Rasch analyses is warranted.

Response: The details of questionnaire items and subscales have been provided in Table 1 (Page 8). 'There are various schools of thought regarding sample size requirements for validation studies. We aim to recruit at least 180 patients based on the recommendations found in contemporary literature outlining psychometric best-practice (5 to 10 times the number of variables in any given multivariate statistical model).' (Pages 29 - 32)

14. Abstract: spell out PRO-trACK acronym

Response: This has now been written in full. See introductory paragraph of abstract.

15. On page 15, it states that patients will be "randomly enrolled into groups of 3-5 patients per group." Consider a different word choice for "randomly," which suggests randomization.

Response: We understand the reviewer's point. The sentence has now been rephrased. 'The process of improving the usability of any system is an iterative one; therefore a minimum of 2 testing cycles will be conducted with the patients'. (Page 15, paragraph 1)

16. On page 20, there is a disclaimer statement for patients that results will not be fed back to clinicians for Aim 3. Isn't this true for aims 1-2 too?

Response: We understand the reviewer's point but we decided to emphasize this disclaimer for the validation study as this is the only time patients will remotely complete the questionnaires based on their actual health status. The patients who participate in the usability testing will be completing the questionnaires within a pre-defined scenario and the aim is purely to receive feedback on the user-friendliness of the ePROM system. The face-to face interviews will only explore patient symptoms in general terms and if any serious issues are raised patients would be directed to seek appropriate help. As the validation study will be conducted remotely, patients might wrongly assume that clinicians will be able to view their results.

Reviewer 3 - Helen Noble

We are grateful for the comments of Dr Noble and have addressed them below

1. Abstract and main text: I am not sure why you call this a project made up of 3 studies. It appears to be a research study made up of 3 parts.

Response: We thank the reviewer for the comment. We have chosen to use the term project as the ePROM system will be eventually implemented for use at the Renal Unit, Queen Elizabeth Hospital, Birmingham. Although the components of the project are related, they are separate studies and will be written and published separately.

2. I suggest under dissemination you mention international dissemination pg 4 - strengths and limitations. Why is the ePROM only intended for stage 4 & 5 CKD patients?

Response: A sentence has been added to the dissemination section. 'We will also disseminate our findings at seminars and conferences both nationally and internationally'. (Page 34, under dissemination)

'We have selected this group of patients (stage 4 & 5 CKD) as our main target for this project because, even though they have not yet reached ESRD, they are likely to have high symptom burden and a high risk of rapid clinical deterioration to ESRD. This group of patients are therefore likely to derive the most benefit from an ePROM system. The research team also is working on a related project focused on dialysis patients.' (Page 11, paragraph 1)

3. Top of pg 8 - one study in 3 parts?

Response: Please see response to comment 1.

4. Pg 11- line 49. Something brief about the renal risk calculator would be of interest to readers.

Response: We thank the reviewer for this comment. A sentence has been added to the manuscript. (Page 11, paragraph 1)

5. Line 54 - requires reference. I haven't come across the statement that ESRD is when eGFR falls below 10.

Response: We thank the reviewer for this comment. The statement has been clarified.

'We will recruit adult patients with advanced CKD stages 4 and 5 under the care of the renal services at QEHB, with eGFR < 20ml/min/1.73m² who have been counselled about treatment modalities for end-stage renal disease (ESRD). In addition, using the renal risk calculator, they must have a >20% projected risk of progressing to ESRD and requiring RRT or an eGFR <10ml/min/1.73m² within 2-years.' (Page 10)

6. pg 12 - line 5 - be more tentative.....they don't always have the highest symptom burden

Response: The phrase has been amended to read 'they are likely to have high symptom burden'. (Page 11, paragraph 1)

7. pg 329 - how much funding is supporting the study

Response: This is a PhD studentship funded by the Health Foundation. See page 35.